# Job Exposure Matrix for Chrysotile Asbestos Fibre in the Asbestos Cement Manufacturing (ACM) Industry in Zimbabwe

**DOI:** 10.3390/ijerph19052680

**Published:** 2022-02-25

**Authors:** Benjamin Mutetwa, Dingani Moyo, Derk Brouwer

**Affiliations:** 1Faculty of Health Sciences School of Public Health, University of the Witwatersrand, Johannesburg 2193, South Africa; moyod@iwayafrica.co.zw (D.M.); derk.brouwer@wits.ac.za (D.B.); 2Faculty of Medicine and Health Sciences, Midland State University, Gweru 054, Zimbabwe; 3Department of Community Medicine, Faculty of Medicine, National University of Science and Technology, Bulawayo 029, Zimbabwe

**Keywords:** job exposure matrix, chrysotile asbestos, asbestos-related disease, occupational exposure

## Abstract

Occupational chrysotile asbestos exposure data in Zimbabwe is limited. The aim of this study was therefore to develop a job exposure matrix (JEM) specific to the chrysotile asbestos cement manufacturing industry using the available personal exposure concentration data. Quantitative personal exposure chrysotile fibre concentration data collected by the two factories from 1996 to 2020 were used to construct the JEM. Exposure groups from which data was extracted were classified based on the Zimbabwe Standard Classification of Occupations (ZSCO), 2009–2019. Analysis of amphiboles in raw chrysotile was done by scanning electron microscopy (SEM) and energy dispersive spectroscopy (EDS). Descriptive statistics, namely mean, standard deviation and range were computed for the main variable, job/occupation. All jobs/occupations in both factories had annual mean personal exposure concentrations exceeding the OEL of 0.1 f/mL, except for the period from 2009 to 2016 in the Harare factory and the period from 2009 to 2020 in the Bulawayo factory. Despite the Harare factory having no AC manufacturing activity since 2017, personal exposure concentrations showed elevated levels for the period 2018–2020. Amphiboles were detected in almost all bulk samples of chrysotile asbestos analysed. The established JEM, which has been generated from actual local quantitative exposure measurements, can be used in evaluating historical exposure to chrysotile asbestos fibre, to better understand and predict occurrence of ARDs in future.

## 1. Introduction

Asbestos is a group of naturally occurring fibrous silicate minerals that include chrysotile, crocidolite, amosite, anthophyllite, tremolite and actinolite [1,2,3,4]. These fibrous materials are resistant to heat, fire and corrosion, extremely durable and because of such properties, they have found widespread use in industry [2,3,4,5]. Today, Russia, China, Brazil and Kazakhstan are leading producers of chrysotile asbestos. Zimbabwe has been a major producer of chrysotile asbestos; however, full-scale mining of the mineral ceased in 2010. The major consumer of chrysotile asbestos was the asbestos cement manufacturing (ACM) industry, taking up about 10% of the produced chrysotile asbestos, while 90% was exported. Since 2010, chrysotile mainly used in the ACM industry in Zimbabwe has been largely imported from Russia. Currently the chrysotile mines are harnessing chrysotile from chrysotile dumps, and there are efforts by government to resume full-scale mining of chrysotile, making Zimbabwe the only country in Africa to still be producing and using chrysotile asbestos. 

In Africa, major producers of asbestos were South Africa, Swaziland, and Zimbabwe [6]. Production of chrysotile asbestos was about 17,000 metric tonnes (mt), rising to about 50,800 mt in 1940 and reaching a peak of 250,949 mt in 1980. By 2010, production dropped to 2400 mt. 

Occupational exposure to all forms of asbestos, including chrysotile, have been associated with risk of asbestos-related diseases (ARDs), such as lung cancer, mesothelioma and cancer of the larynx and ovary [1,2,4,7]. ARDs have been observed to have a dose response relationship with a long latency period between exposure and onset of disease. The minimum latency period generally associated with onset of most ARDs is 10 years depending on levels of exposure. Hence the estimation of past exposures before occurrence of ARD is crucial to elucidate the association between occupational exposure and onset of disease [3,4,8]. Occurrence of ARDs generally can be determined by the historical exposure to asbestos of the individual affected by the ARD [1,9].

Job exposure matrices (JEMs) have been used as tools for assessing past exposure levels to various hazardous factors. Historical exposure to workplace hazards and indeed chrysotile asbestos is a key factor in the onset of ARDs in Zimbabwe because chrysotile asbestos has been used in manufacturing asbestos cement (AC) products in construction works since the 1940s.

The principle of JEMs is based on the construction of a database that associate exposures to various hazardous factors with occupations/jobs or workstations [4,10,11]. Thus, a JEM is a tool through which information on jobs collected in epidemiological studies may be converted into information on possible exposures [12]. Essentially, the key objective of a JEM is to try and link job/occupation information with workplace hazardous exposure information. The idea of a JEM dates back to the time when Ramazzini tried to link diseases in 52 occupations to which the occupations were exposed to the respective hazards. In 1941, the first JEM to be developed consisted of a cross tabulation of an occupations list with that of a list of hazards [13]. Hence, the concept of JEM is that it is essentially a table in which one axis is comprised of occupations/jobs, while the other axis is comprised of workplace hazards. Additionally, for a given job/occupation each cell of the matrix can contain qualitative or quantitative exposure indicators. The JEMs may be constituted by four axes namely job/occupation, agent of exposure, time or time-period and place/location [12,14]. Exposure can vary with respect to occupations/jobs and workplaces and thus jobs can be categorised into homogenous groups to reflect similar exposures. Hence, workers exposed to a particular agent under similar or same conditions should correspond to the same entry of the matrix [14]. Furthermore, JEMs for application in retrospective studies should consider changes in exposure over time to aid in assigning health outcomes at a point in time in future. In this respect a time variable must be introduced when exposure has changed over time [12,14]. It is also important to include the place/location variable in JEMs since exposure may vary across different plants or factory situated in different locations [12].

Quantitative exposure measurements have often been considered as best estimates of actual dose [15,16]. Hence the JEM provides possible dose estimates for use in dose-response relationship studies. Where measurement data was available, it has been used in the development of JEM in workplace settings [12,17]. 

Information sources for which exposure estimates for a JEM can be obtained include actual measurements collected over time in workplace plants or factories of interest, company occupational hygienists, scientific literature and exposure data banks [14,18]. It is important to note that, in this study, data collected spanning almost two and half decades provided a good resource to obtain exposure estimates upon which the JEM was built.

JEMs have some limitations, among them being that variability of exposure within occupational or job classes in different workplaces, countries or over time are usually not considered in applying the JEM, leading to possible exposure misclassifications [19]. Despite some limitations, the JEM approach has advantages that can be used in situations in which traditional methods for occupational exposure assessment may be difficult or impossible to implement [10]. Additionally, JEM have become favoured approaches for occupational exposure assessment in industrial cohort studies of cancer. They are also commonly used as common occupational hygiene tools applied for accident prevention in the workplace. JEMs have also been used extensively in industry-specific studies for various study designs to aid in the retrospective evaluation of occupational exposure in employees whose exposure history may not be readily available [14].

A chrysotile asbestos JEM built using historical exposure is important to aid in the prevention and prediction of occupational cancers with long latency periods. Nonetheless, to the best of our knowledge, there are no JEMs developed for various workplace hazards in Zimbabwe industry sectors and in particular a JEM focusing on chrysotile ACM industry; thus, this is the first one of its kind in a Zimbabwe workplace setting. A JEM specific for chrysotile asbestos in ACM industries in Zimbabwe will be useful in future epidemiology studies rather than using or extrapolating exposure estimates from international studies which may not be suitable for Zimbabwe workplaces settings. Hence, this study aimed to construct a JEM using quantitative occupational exposure data produced by the ACM industry over a period of about two and half decades and qualitative information on possible amphibole presence in the chrysotile asbestos being used in the manufacture of AC products.

## 2. Materials and Methods

Personal exposure data measured for the period 1996 to 2020 extracted from paper records of the two main manufacturing factories in Harare and Bulawayo cities were used to build the JEM. Harare is the capital city of Zimbabwe, in the northern part of the country, 387 km from the town of Zvishavane, while Bulawayo is the second largest city and is situated in the southern part of the country, 184 km from the town of Zvishavane, where chrysotile asbestos mines Shabanie and Mashava are located. The data was comprised of all personal exposure measurements collected by the company for close to 25 years in various operational areas examined in the two AC manufacturing factories. The data collected was examined to assess the chrysotile asbestos exposure for each combination of job, time period, place and mean personal exposure level and possible amphibole contamination. The industry is the chrysotile asbestos cement manufacturing. The jobs were classified into 9 broad similar or homogenous categories, namely saws cutting, fettling, moulded goods, kollergang, ground hard waste, laundry, pipe joints and multi-cutter operators.

Measurement of airborne chrysotile asbestos followed the standard method of the Asbestos International Association (AIA) Reference method for the determination of airborne asbestos fibre concentrations at workplaces by light microscopy as previously described by Mutetwa et al. (2021) [20]. Briefly, the chrysotile fibres were sampled on 25 mm membrane filters of 1.2-µm pore size with printed grids and then counted by means of a phase contrast microscope (PCM). The fibres counted were generally longer than 5 µm with a width of less than 3 µm and length-to-width ratio of more than 3:1.

Detection of amphiboles in the chrysotile asbestos being used for manufacturing AC products in the Bulawayo factory was done by National Institute of Occupational Health (NIOH)—National Health Laboratory Service (NHLS), South Africa, using scanning electron microscopy (SEM) and energy dispersive spectroscopy (EDS).

### 2.1. Data Analysis

Data analysis was conducted using IBM SPSS version 26. Monthly averaged personal exposure concentrations for the factories were used. Mean personal exposure concentrations were analysed per operator working in a particular location per factory.

ANOVA was applied with the aim of identifying patterns of exposure variability among the time-period for various job categories and determining whether there was a statistically significant difference in exposure concentration between the four time-periods for various jobs. A Tukey post hoc test (Tukey’s honest significance difference test) was run to find out which specific group means of time-periods for various jobs/occupations (compared with each other) were different. 

The arithmetic mean was used as a representative value for analysis of the measurements as this is normally taken as the best summary measure of exposure in epidemiological studies of chronic diseases when adopting a linear exposure response model [7,21].

### 2.2. Ethics

The study was approved by the University of the Witwatersrand Human Research Ethics Committee (clearance certificate number M181157) and the Medical Research Council of Zimbabwe (MRCZ) (approval number MRCZ/A/2445).

## 3. Results

A total of 3066 airborne chrysotile personal measurements collected from company records, spanning a period of about 25 years from which 1788 annual mean personal exposure concentrations were drawn from, were used to build the job exposure matrix (JEM) for chrysotile asbestos fibre in the AC manufacturing factories. For the purpose of the JEM, jobs selected had the most data and were in most common operational areas even up to 2020. The jobs involved are as outlined in Table 1, and their description is briefly given. Additionally, the jobs were coded with respect to the Zimbabwe Standard Classification of Occupations (ZSCO) [22].

Table 1 presents the JEM with jobs categorised with their description, factory location, mean and range, period and the possible amphiboles identified in chrysotile asbestos materials used in the manufacturing process. Table 2 shows statistical significance in variability in exposure concentrations for time periods for various job categories. Appendix A further show post hoc output showing statistical significance in exposure concentrations for various time periods for each job category. 

Table 3 shows results of type of amphiboles detected in the bulk chrysotile asbestos samples collected in bags of asbestos used in the manufacturing process in the Bulawayo factory.

Annual mean personal exposure concentration for saw cutting, fettling, ground hard waste operators (Harare and Bulawayo factory), laundry room operator, moulded goods operator (Harare factory), and pipe section operators (Bulawayo factory) showed high levels exceeding the OEL of 0.1 f/mL for the time-period 1996 to 2008. As reported by Mutetwa et al. (2021), high exposure levels above the OEL were exhibited in the 1990s and 2000s compared to the period 2009 to 2016 for both factory locations with saw cutting, kollergang and ground hard waste operators in both locations generally exposed to high levels of airborne chrysotile fibre in the years 1996 to 2000 [20]. It is insightful to note that the results for the Harare factory for the years 2018–2020 for all operational areas had exposure concentrations exceeding the OEL, even though manufacturing of AC products during this period had ceased; however, exposure concentrations were also comparable to those reported previously by Mutetwa et al. (2021), for the period 1996 to 2008. The results for the measurement period of 2017–2019 for the Bulawayo factory were all below the OEL for all key operational areas examined, although they were lower than the exposure concentrations reported by Mutetwa et al. (2021) [20], for the period 2009 to 2016, though the order of magnitude is about the same.

The factories reported that importation of chrysotile asbestos started in 2008 following marked decline of mining operations at the two chrysotile mines of Shabanie and Mashava and eventual ceasing of mining operations in 2010. This would suggest that for the Harare factory manufacturing of AC products continued for 8 years with the use of imported fibre, while for the Bulawayo factory, manufacturing of AC products using imported fibre has been ongoing for 12 years up to 2020. However, manufacturing at the Bulawayo factory continues to this day, using largely imported fibre and a small component from locally produced fibre harnessed from chrysotile dumps.

For the Harare factory, during the period from 2017 to 2020, operations in areas where personal samples were collected, the jobs involved essentially care, maintenance and general cleaning of equipment, except for the saws cutting where cutting of AC sheets from the Bulawayo factory continued as what used to happen during the time manufacturing was ongoing for the period 1996 to 2016.

For the 6 locally produced chrysotile samples analysed, 4 had amphiboles detected, namely tremolite and anthophyllite as shown in Table 3. Furthermore, for the imported chrysotile asbestos, amphiboles, namely tremolite, crocidolite and actinolite were also detected in 5 out of 6 samples analysed (Table 3).

Variability of mean personal exposure concentrations between time periods for each job category was also tested using ANOVA. There was a statistically significant difference in the annual mean personal chrysotile exposure concentrations among the different time periods for various job categories as determined by one-way ANOVA in both factory locations except for the fettling table operator in Harare (Table 2).

Furthermore, for the Harare factory after 2016, jobs in the areas examined essentially had similar activities of care and maintenance of equipment and cleaning in the respective areas which involved AC manufacturing and handling, and despite no manufacturing of AC products taking place, exposure concentrations remained elevated above the OEL of 0.1 f/mL, except for saw cutting operator, in which exposure concentration was same as the OEL. Additionally, the exposure concentrations for the time period 2018 to 2020 were statistically significantly higher compared to the prior period of 2009 to 2016 (*p* < 0.001).

For almost all jobs in the Harare factory, the post hoc tests show that exposure concentrations during the period 2009 to 2016 was statistically significantly lower than exposure concentrations during the periods 1996–2000, 2001–2008 and 2018–2020. Additionally, for the laundry operator, post hoc test further reveals that exposure concentrations during the period 2018–2020, although elevated above the OEL of 0.1 f/mL, was statistically significantly lower and higher than exposure concentrations during the periods, 1996-2000 and 2009–2016, respectively. There was no statistically significant difference in exposure concentrations between the time periods 1996–2000 and 2018–2020 for all jobs (*p* > 0.05) except for the saw cutting operator (*p* < 0.05) in the Harare factory (Appendix A).

For the Bulawayo factory, the post hoc test (Appendix A) shows that exposure concentrations during the period 1996 to 2000 was statistically significantly higher than exposure concentration during the periods 2009–2016 and 2017–2019, with saw cutting, kollergang and pipe joints operators exposure concentrations also being statistically significantly higher during the period 1996–2000 than during the period 2001 to 2008 (*p* < 0.05). There was no statistically significant difference in exposure concentrations between the time periods 2009–2016 and 2017–2019 for all jobs (*p* > 0.05).

### Analysis of Presence of Amphiboles in Samples Collected from Bags of Raw Chrysotile Materials Used for Manufacturing AC Products in the Bulawayo Factory

Six (6) local bulk chrysotile samples and 6 imported chrysotile samples were randomly collected from bags ready to be processed at the holding bay and at the kollergang area. Fibres with aspect ratio greater than 3:1 were observed in all the samples morphologically resembling asbestos using SEM.

## 4. Discussion

The focus of this study was on the construction of a JEM using a large number of personal chrysotile asbestos fibre measurements relevant to the Zimbabwean AC industry and collected over a period of about 25 years. The data used were related to work characteristics of jobs outlined under the results section, which jobs are the most common jobs in the ACM industry in Zimbabwe and the matrix arising therein provides a tool for exposure assessment in future studies.

The job categories with high exposure levels were saw cutting, fettling, ground hard waste, laundry room and multi-cutter operator and such levels of exposure may present increased risk of ARDs. As reported by Mutetwa et al. (2021) [20], exposure concentrations declined over time for both factories for the period 1996 to 2016, and further declines in exposure concentrations for all jobs in the Bulawayo factory was observed for the period 2017 to 2019. Exposure concentrations in the Harare factory, for the period 2018 to 2020 were, however, much higher than those reported by Mutetwa et al. (2021) [20], for the preceding time period 2009 to 2016, during which concentrations ranged from 0.05 to 0.07 f/mL in various operations examined compared to 0.10 to 0.12 f/mL for the period of 2018 to 2020, even though manufacturing had ceased. The Harare factory reported no clean-up before the current activities of manufacturing concrete tiles and other concrete products started, so resuspension of chrysotile fibre from the floors which accumulated chrysotile fibre in the past could be responsible for the elevated levels for the time-period 2018 to 2020. The work practices deployed during the period for manufacturing of AC products, e.g., wet dust suppression methods may no longer be practiced.

On the other hand, the lower exposure concentration values exhibited in the Bulawayo factory for the period 2017 to 2019 compared to 2009 to 2016 fibre concentration levels reported by Mutetwa et al. (2001) [20] may suggest continued adherence to good work practices and continued implementation of occupational safety and health management systems which the factory has been subscribing to over the years in its AC manufacturing processes. Although the factories were of the same company, it can be viewed that since the Harare factory was now producing concrete products, exposure to chrysotile asbestos fibre could have been considered not much of a threat to health, hence the low OSH standards at the Harare factory compared to the Bulawayo factory.

During the period 1996 to 2000, the chrysotile ACM industry developed and used its own occupational exposure standard of 0.2 f/mL in the absence of a national statutory limit. This may mean that exposure concentrations up to this limit were deemed as presenting insignificant health risk to workers exposed, hence control measures were then possibly designed to contain airborne chrysotile levels to be within 0.2 f/mL. However, exposure at a level of 0.2 f/mL does present some health risk of ARDs, in light of the fact that the occupational exposure limit in many countries, including Zimbabwe, has been set at 0.1 f/mL [23,24,25], as an attempt to minimise health risks associated with exposure to asbestos.

The categorisation of exposure measurements into measurement periods reflecting ACM industry occupational hygiene practices as well as general economic status provides insights into variation of exposure estimates over time [16]. Post hoc test data further demonstrate that the earlier time periods of 1996–2000 had statistically significantly higher exposure concentrations than exposure concentrations during the period 2001–2008 and 2009–2016 in both factories (*p* < 0.05). Such high exposure concentrations, as reported by Mutetwa et al., 2021 [20], during the 1990s and early to mid-2000s as well as the industry’s own belief that maintaining exposure to below 0.2 f/mL may be the reason for higher exposure levels exhibited in the 1990s and early 2000s compared to periods 2009–2016.

The detection of amphiboles in the chrysotile asbestos being used in the manufacture of AC products further heighten the possible risk of ARD occurrence among workers exposed. Anthophyllite fibres have also been detected in the past by XRD and TEM in Zimbabwe chrysotile [26,27]. Amphiboles, particularly crocidolite have significantly been associated with the development ARDs, such as mesothelioma and lung cancer [1,2,28]. Significant importation of chrysotile asbestos has been ongoing since 2008 and the detection of amphiboles and notably crocidolite in some of the samples taken from the chrysotile bags further suggests a serious risk of ARD occurrence in the form of mesothelioma as crocidolite is one the most dangerous form of asbestos [28]. The Bulawayo factory reported that about 5200 tonnes per annum of chrysotile is imported and 550 tonnes annually of local chrysotile is used for manufacturing AC products. This translates to about 69,000 tonnes and 6600 tonnes of imported and local chrysotile asbestos, respectively, being consumed for the 12-year period to 2019 from the time significant imports started being used in 2008. Assuming that the Harare factory also used similar quantities of asbestos per annum, for the 8-year period to 2016 before AC production ceased, a considerable amount of chrysotile asbestos amounting to 46,000 tonnes (41,600 imported plus 4400 tonnes local was used up to the time AC manufacturing ceased. While chrysotile asbestos is dangerous to human health, the possible presence of amphiboles in the chrysotile being used could further amplify the health risk presented by exposure to asbestos in these factories.

Kollergang operators who handle raw chrysotile asbestos, ground hard waste operators who are involved in a hazardous process that generate considerable dust in both factories particularly during the 1996 to 2000 and 2001 to 2008 time periods were possibly at high risk of exposure to amphiboles contaminants in chrysotile asbestos being used in manufacturing of AC products. Additionally, higher exposure concentrations in lathe machines operators in the Bulawayo factory may also suggest elevated risk of exposure to amphiboles during the earlier time periods of 1996 to 2008 and thus possible increased risk of ARDs for operators working with these machines.

### Strength and Limitations

The strength of the JEM is that the data used in its construction is purely from the local ACM industry workplace settings. It can also be useful in the evaluation of the contribution of asbestos exposure on ARD occurrence, taking into account the job profile of exposed workers and time period. Fewer measurements data for the period 2016 to 2020 diminished the accuracy of the mean exposure concentration for this time period. “Additionally, while phase contrast microscopy (PCM) provides relatively quick and cost-effective analysis of asbestos samples, the PCM is not able to distinguish whether fibres observed are chrysotile or amphiboles fibres. Nonetheless, in this study, it was considered that fibres were generally chrysotile as the factories have been using chrysotile asbestos in their manufacturing processes since their establishment in the 1940s”. In view of limited resources, the factories were not able to participate in interlaboratory quality assurance and control fibre counting programmes, particularly for the later years of 2013 to 2020; hence, this could have presented a limitation in the results. The factories, however, continued to consistently apply the standard method for asbestos measurements as previously reported by Mutetwa et al., (2021), making sure that use of blank samples was always part of the methodology.

## 5. Conclusions

The JEM generated in this study provides quantitative estimates of personal chrysotile asbestos exposure concentrations for workers operating in ACM plants, based on jobs held and factory locations worked in and may give estimates of latency based on estimates of time period of exposure used. Furthermore, the JEM may provide an opportunity for prediction of occurrence of ARDs and possible analysis of exposure response relationships that may be linked to exposure episodes of measurement period in the distant past. The JEM also gives a perspective on the possible amphiboles associated with the local and imported chrysotile asbestos used in the manufacturing processes.

## Figures and Tables

**Table 1 ijerph-19-02680-t001:** Job exposure matrix based on airborne chrysotile asbestos fibre occupational exposure data for ACM industry factories for the period 1996–2020. The codes in brackets in the Job column is the ZSCO code.

Job	Job Description	Time Period	Harare Factory (Tr, Anth, Cr, Act)				*p*-Value	Bulawayo Factory (Tr, Anth, Cr, Act)				*p*-Value
			N	Mean ± SD	95%CI	Range		N	Mean ± SD	95%CI	Range	
				(f/mL)	LB UB	Min Max			(f/mL)	LB UB	Min Max	
Saw cutting operator (1023)	Cutting by saw asbestos sheets and facia boards to size	1996–2000 2001–2008 2009–2016 2018–2020 *	60 88 77 29	0.19 ± 0.01 0.13 ± 0.02 0.07 ± 0.02 0.10 ± 0.02	0.19 0.19 0.12 0.13 0.07 0.08 0.09 0.11	0.16 0.24 0.08 0.18 0.03 0.11 0.06 0.15	<0.001	50 49 14 24	0.17 ± 0.02 0.12 ± 0.02 0.06 ± 0.02 0.05 ± 0.01	0.16 0.18 0.11 0.12 0.05 0.07 0.05 0.06	0.12 0.24 0.09 0.16 0.01 0.08 0.05 0.07	<0.001
Fettling table operator (1023)	Scrapping/polishing AC moulded goods	1996–2000 2001–2008 2009–2016 2018–2020 *	53 73 Nil 4	0.12 ± 04 0.12 ± 02 - 0.11 ± 03	0.11 0.13 0.12 0.13 - 0.05 0.16	0.05 0.18 0.05 0.19 - 0.06 0.14	0.561	40 11 - -	0.17 ± 0.06 0.12 ± 0.03 - -	0.16 0.19 0.10 0.14 - -	0.07 0.30 0.06 0.15 - -	<0.001
Moulded goods operator (1024)	Moulding of AC goods under wet conditions	1996–2000 2001–2008 2009–2016 2018–2020 *	58 82 52 5	0.11 ± 0.04 0.11 ± 0.04 0.05 ± 0.01 0.11 ± 0.02	0.10 0.12 0.11 0.12 0.05 0.06 0.08 0.13	0.04 0.20 0.03 0.18 0.03 0.08 0.08 0.13	<0.001	-	-	-	-	-
Kollergang operator (1021)	Opening of & loading chrysotile bags into process machine and operate machine	1996–2000 2001–2008 2009–2016 2018–2020 *	58 81 64 9	0.13 ± 0.04 0.12 ± 0.02 0.07 ± 0.02 0.12 ± 0.01	0.12 0.14 0.11 0.12 0.06 0.07 0.11 0.13	0.05 0.20 0.04 0.16 0.04 0.11 0.10 0.13	<0.001	36 42 33 15	0.14 ± 0.03 0.12 ± 0.01 0.07 ± 0.03 0.06 ± 0.01	0.13 0.15 0.11 0.12 0.06 0.08 0.05 0.07	0.08 0.24 0.08 0.14 0.03 0.18 0.03 0.09	<0.001
Ground hard waste operator (1021)	Feeding AC waste materials into grinder machine	1996–2000 2001–2008 2009–2016 2018–2020 *	57 56 55 8	0.16 ± 0.03 0.13 ± 0.03 0.07 ± 0.02 0.12 ± 0.01	0.15 0.16 0.12 0.14 0.06 0.08 0.12 0.13	0.08 0.22 0.03 0.20 0.02 0.17 0.11 0.13	<0.001	44 15 5 12	0.13 ± 0.04 0.11 ± 0.04 0.07 ± 0.02 0.06 ± 0.02	0.11 0.14 0.10 0.11 0.05 0.09 0.05 0.06	0.07 0.24 0.08 0.13 0.04 0.09 0.04 0.08	<0.001
Laundry room operator (1024)	Laundering of PPC using wash machine	1996–2000 2001–2008 2009–2016 2018–2020 *	47 87 15 14	0.13 ± 0.03 0.13 ± 0.02 0.05 ± 0.01 0.11 ± 0.02	0.12 0.14 0.12 0.13 0.04 0.05 0.10 0.12	0.06 0.20 0.06 0.21 0.03 0.07 0.07 0.14	<0.001		-	-	-	-
Pipe joints operator (1023)	Lathe machining of AC joints pipes	1996–2000 2001–2008 2009–2016 2018–2020	-	-	-	-	-	44 46 9 4	0.13 ± 0.04 0.11 ± 0.01 0.05 ± 0.02 0.05 ± 0.02	0.12 0.14 0.11 0.12 0.05 0.07 0.02 0.08	0.06 0.30 0.08 0.15 0.04 0.08 0.02 0.08	<0.001
Full length pipe operator (1023)	Lathe machining & polishing of full-length AC pipe joints	1996–2000 2001–2008 2009–2016 2018–2020	-	-	-	-	-	43 45 9 -	0.13 ± 0.04 0.11 ± 0.01 0.07 ± 0.02 -	0.12 0.14 0.11 0.11 0.05 0.08 -	0.06 0.27 0.07 0.14 0.04 0.08 -	<0.001
Multi-cutter operator (1023)	Cutting full length pipes into collars for coupling pipes	1996–2000 2001–2008 2009–2016 2018–2020	-	-	-	-	-	26 36 2 2	0.13 ± 0.04 0.12 ± 0.01 0.07 ± 0.03 0.04 ± 0.01	0.11 0.14 0.12 0.13 0.12 0.13 0.00 0.11	0.05 0.20 0.10 0.14 0.05 0.20 0.04 0.05	<0.001

ACM—asbestos cement manufacturing, AC—asbestos cement, PPC—personal protective clothing, SD—standard deviation, Min—minimum, Max—maximum, N—number of monthly-averaged personal chrysotile fibre concentrations, 1996—2020, LB—lower bound and UB—upper bound values for the 95% confidence intervals of the mean. () Bracketed number refers to Zimbabwe Standard Classification of Occupations code (ZSCO) (NSSA, 2009—2019), Tr—tremolite, Anth—anthophyllite, Cr—crocidolite, Act—actinolite. * Care and maintenance of equipment and cleaning—Harare factory when manufacturing of AC products no longer takes place. ** Possible exposure to amphiboles in both factories could have started in 2010 following major shift in use of imported fibre.

**Table 2 ijerph-19-02680-t002:** Analysis of variability of mean personal exposure concentrations between time period catergories for various jobs.

	Harare Factory			Bulawayo Factory		
Operator	df	F	*p*-Value	df	F	*p*-Value
Saw cutting operator	2, 250	519.6	<0.001	3, 134	236.8	<0.001
Fettling table operator	2, 127	0.6	0.561	1, 49	5.2	<0.001
Moulded goods operator	3, 193	59.2	<0.001	-	-	-
Kollrgang operator	3, 208	83.3	<0.001	3, 123	68.9	<0.001
Ground hard waste	3, 172	96.9	<0.001	3, 73	18.2	<0.001
Laundry	3, 169	48.0	<0.001	2, 26	5.0	<0.001
Pipe joints	-	-	-	3, 100	24.8	<0.001
Full-length	-	-	-	2, 26	20.6	<0.001
Multicutter	-	-	-	3, 62	9.3	<0.001

Df—degrees of freedom. F—F test statistic.

**Table 3 ijerph-19-02680-t003:** Amphiboles in chrysotile samples collected from bags of raw chrysotile material used for manufacturing AC products.

Chrysotile Sample	EDS Result
Local 01	Straight and curved fibres exhibited peaks of magnesium and silicon Chrysotile and tremolite detected in sample
Local 02	Straight and curved fibres exhibited peaks of magnesium and silicon. Chrysotile, tremolite and anthophyllite detected in the sample
Local 03	Straight and curved fibres exhibited peaks of magnesium and silicon. Chrysotile, tremolite and anthophyllite detected in the sample
Local 04	Curved fibres exhibited peaks of magnesium and silicon Chrysotile only detected in the sample
Local 05	Curved fibres exhibited peaks of magnesium and silicon. Chrysotile only detected in the sample
Local 06	Curved fibres exhibited peaks of magnesium and silicon Chrysotile and tremolite detected in the sample
Imported 01	Curved fibres exhibited peaks of magnesium and silicon. Chrysotile and tremolite detected in the sample
Imported 02	Curved and straight fibres exhibited peaks of magnesium and silicon. Chrysotile, crocidolite and tremolite detected in the sample.
Imported 03	Curved and straight fibres exhibited peaks of magnesium and silicon. Chrysotile, crocidolite, tremolite and actinolite detected in the sample
Imported 04	Curved and straight fibres exhibited peaks of magnesium and silicon. Chrysotile, tremolite and actinolite detected in the sample
Imported 05	Curved fibres exhibited peaks of magnesium and silicon. Chrysotile only detected in the sample
Imported 06	Curved fibres exhibited peaks of magnesium and silicon. Chrysotile and tremolite detected in the sample

## Data Availability

The dataset used in this study are available from the corresponding author on reasonable request. The datasets are not publicly available to maintain confidentiality of the factories used in the study.

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
