# Peer review of "Job Exposure Matrix for Chrysotile Asbestos Fibre in the Asbestos Cement Manufacturing (ACM) Industry in Zimbabwe"

_ijerph, 2022, doi:10.3390/ijerph19052680_

Round 1
Reviewer 1 Report
Dear authors,
I have read your manuscript with interest. I do believe that this study provides valuable data combining fibre measurement at these asbestos-cement manufactoring industries in Zimbabwe with main jobs. I agree that this JEM could be useful for future studies on ARD in Zimbabwe.
My questions are mainly related to the methods used to mesure asbestos in the air samples. Authors mention that light microscopy was used. As it is already known, light microscopy has a limited sensitivity for detecting asbestos fibres and this limitations shoud be mentioned and discussed. Moreover, how were samples analyzed to ascertain the type of asbestos? Authors shoud mention the method used, for example energy-dispersive x-ray analysis or whatever alternative method. This aplies for determining chrisotyle asbestos and also amphyboles (local bulk samples).
I have had some problems to understand properly Table 1. When displaying f/ml values, minimum and maximal values are misleading (for example, for saw cutting in Harare minimal value is 0,19 and maximal is 0,16). Additionally, the meaning of "LB" and "UB" should be exposed in the footnote.
The same lack of explanatory footnote is observed in Table 2
In the discussion, the higher fibre values in 2018-2020 in respect to 2009-2016 in Harare is attributed to the lack of cleaning-up. However, if this had been the problem, the same lack of cleaning up would had happened during 2009-2016, when lower levels were detected.
I think authors should improve their discussion in page 9. The improvement observed in Bulawayo during 2017-2019 is attributed to adherence to good work practices and I do not understand why the same company had different work practices in the different settlements.
Authors state that the risk of developing ARD is real if materials are contaminated with amphiboles. However, there is no doubt that chrysotile fibers are also dangerous and I do believe that this point should be clearly stated in the manuscript.
Taking into account the abovementioned, I think authors should list some other limitations. The reduced number of measurements during 2016-2020 should be previously mentioned in methods, otherwise the reader assumes that the number of measurement were homogeneous across the different time periods.
Reviewer 2 Report
This study uses a job exposure matrix (JEM) to evaluate the historical exposure to asbestos in two Zimbabwean asbestos plants during 15 years. Of particular interest is that the data are specified for different types of common jobs in the industry.
This relatively short manuscript is well structured and written in an accessible way. The tables are useful and do not attract particular comments
Weaknesses of the study are a lack of quality control on the data provided by the industry.
The local character of the study hampers the international relevance and application of the conclusions.
Round 2
Reviewer 1 Report
Dear authors, thank you for your responses. I do not have further comments